# BinGAN: Learning Compact Binary Descriptors with a Regularized GAN

**Maciej Zieba**
Wroclaw University of
Science and Technology, Tooploox
maciej.zieba@pwr.edu.pl

**Piotr Semberecki**
Wroclaw University of
Science and Technology, Tooploox
piotr.semberecki@pwr.edu.pl

**Tarek El-Gaaly**
Voyage
tarek@voyage.auto

**Tomasz Trzcinski**
Warsaw University of Technology,
Tooploox
t.trzcinski@ii.pw.edu.pl

## Abstract

In this paper, we propose a novel regularization method for Generative Adversarial Networks, which allows the model to learn discriminative yet compact binary representations of image patches (*image descriptors*). We employ the dimensionality reduction that takes place in the intermediate layers of the discriminator network and train binarized low-dimensional representation of the penultimate layer to mimic the distribution of the higher-dimensional preceding layers. To achieve this, we introduce two loss terms that aim at: (i) reducing the correlation between the dimensions of the binarized low-dimensional representation of the penultimate layer (*i.e.* maximizing joint entropy) and (ii) propagating the relations between the dimensions in the high-dimensional space to the low-dimensional space. We evaluate the resulting binary image descriptors on two challenging applications, image matching and retrieval, and achieve state-of-the-art results.

## 1   Introduction

Compact binary representations of images are instrumental for a multitude of computer vision applications, including image retrieval, simultaneous localization and mapping, and large-scale 3D reconstruction. Typical approaches to the problem of learning discriminative yet concise representations include supervised machine learning methods such as boosting [27], hashing [8] and, more recently, deep learning [23]. Although unsupervised methods have also been proposed [14, 16, 6], their performance is typically significantly lower than the competing supervised approaches.

The goal of this work is to bridge this performance gap by using an intermediate layer representation of a Generative Adversarial Network (GAN) [9] discriminator as a compact binary image descriptor. Recent studies show the powerful discriminative capabilities of features extracted from the discriminator networks of GANs [19, 21]. With a growing number of hidden units in intermediate layers, the quality of vector representations increase, when applied to both image matching and retrieval. This is why typical approaches make use of high-dimensional intermediate representations to generate image descriptors, therefore leading to high memory footprint and computationally expensive matching. We address this shortcoming and build low-dimensional compact descriptors by training GAN with a novel Distance Matching Regularizer (DMR). This regularizer is responsible for propagating the Hamming distances between binary vectors in high-dimensional feature spaces of intermediate discriminator layers to the compact feature space of the low-dimensional deeper layers in the same network. More precisely, our proposed method allows to regularize the output of an intermediate

layer (with low number units) of the discriminator with the help of the output of previous layers (with high number of units). This is achieved by propagating the correlations between sample pairs of representations between the layers. Moreover, to better allocate the capacity of the low-dimensional feature representation we extend our model with an adjusted version of the Binarization Representation Entropy (BRE) Regularizer [5]. This regularization technique was initially applied to increase the diversity of intermediate layers of the discriminator by maximizing the joint entropy of the binarized outputs of the layers. We adjust this regularization term so that it concentrates on increasing the entropy of the particular pairs of binary vectors that are not correlated in high-dimensional space. As a consequence, we keep the balance between propagating the Hamming distances between the layers for correlated vectors and increasing the diversity of the binary vectors in the low-dimensional feature space.

The main contributions of this paper are two-fold. Firstly, we build a powerful yet compact binary image descriptor using a GAN architecture. Secondly, we introduce a novel regularization method that propagates the Hamming distances between correlated pairs of vectors in the high-dimensional features of earlier layers to the low-dimensional binary representation of deeper layers during discriminator training. A binary image descriptor resulting from our approach, dubbed BinGAN, significantly outperforms state-of-the-art methods in two challenging tasks: image matching and image retrieval. Last but not least, we release the code of the method along with the evaluation scripts to enable reproducible research[1].

## 2 Related Work

### 2.1 Binary Descriptors

Binary local feature descriptors have gained a significant amount of attention from the research community, mainly due to their compact nature, efficiency and multitude of applications in computer vision [4, 13, 1, 25, 28, 27, 7]. BRIEF [4], the first widely adopted binary feature descriptor, sparked a new domain of research on feature descriptors that rely on a set of hand-crafted intensity comparisons that are used to generate binary strings. Several follow-up works proposed different sampling strategies, e.g. BRISK [13] and FREAK [1]. Although these approaches offer unprecedented computational efficiency, their performance is highly sensitive to standard image transformation, such as rotation or scaling, as well as other viewpoint changes. To address those limitations, several supervised approaches to learning binary local feature descriptors from the data have been proposed. LDAHash [25] proposed to train discriminative projections of SIFT [17] descriptors and binarize them afterwards to obtain a highly robust patch descriptor. D-Brief [28] extends this approach by increasing the efficiency of the descriptor with banks of simple filtering elements used to approximate the projections. To further boost the performance of learned binary descriptors, BinBoost [27] proposes to use greedy boosting algorithm for training consecutive bits, while RDF [7] uses an alternative greedy algorithm to select the most distinctive receptive fields used to construct dimension of the descriptor. With this kind of approach, it is possible to obtain more powerful descriptors than by application of hand-crafted methods. However, the binary descriptors are trained using pair-wise learning methods, which substantially limits their applicability to new tasks.

### 2.2 Hashing Methods

On the other hand, binary descriptors can be learned with hashing algorithms that aim at preserving original distances between images in binary spaces, such as in [2, 20, 30, 8].

Locality Sensitive Hashing (LSH) [2] binarizes the input by thresholding a low-dimensional representation generated with random projections. Semantic Hashing (SH) [20] achieves the same goal with a multi-layer Restricted Boltzmann Machine. Spectral Hashing (SpeH) [30] exploits spectral graph partitioning to create efficient binary codes. Iterative Quantization (ITQ) [8] uses an iterative approach to find a set of projections that minimize the binarization loss. Unlike most recent deep learning approaches (discussed next), these hashing algorithms typically operate on hand-crafted image representations, *e.g.* SIFT descriptors [25], dramatically reducing their effectiveness and limiting their performance, as can be seen in the results of our experiments.

## 2.3 Deep Learning Approaches

Inspired by the spectacular success of deep neural networks, several methods have been proposed that generate binary image descriptors using deep neural networks [23, 26, 16, 14, 6]. Supervised methods, such as [23, 26], achieve impressive results by exploiting data labeling and training advancements such as Siamese architecture [23] or progressive sampling [26]. Nevertheless, their outstanding performance is often limited to the original task and it is challenging to apply them to other domains.

Unsupervised deep learning methods [16, 14, 6], on the other hand, are typically less domain-specific and do not require any data labeling, which becomes especially important in the domains where such labeling is hard or impossible to obtain, *e.g.* medical imaging. Deep Hashing (DH) [16] uses neural networks to find a binary representation that reduces binarization loss while balancing bit values to maximize its entropy. As an input, however, it takes an intermediate image representation, such as the GIST descriptor. DeepBit [14] addresses this shortcoming by using a convolutional neural network and further improves the results with data augmentation. However, DeepBit relies on a rigid sign function with threshold at zero to binarize the floating-point output values, which may lead to significant quantization losses. DBD-MQ [6] overcomes this limitation by mapping this problem as a multi-quantization task and using K-AutoEncoders network to solve it. In this paper, we follow this line of research and employ a different binarization technique, as in [4], to generate binary descriptors. Furthermore, we also rely on recent generative models, namely Generative Adversarial Networks [9], to build image descriptors. In this regard, our work is also related to [18] and [24], where GANs are used to address image retrieval problem. [18] learns binary representations by training an end-to-end network to distinguish synthetic and real images. [24] proposes to employ GANs to enhance the intermediate representation of the generator. Contrary to our approach, however, both of those methods use tanh-like activation for binarization and optimize their performance toward image retrieval task, while our approach is agnostic to final application and can be equally successful when applied to local feature descriptor learning, image matching or image retrieval.

# 3 BinGAN

We propose a novel approach for learning compact binary descriptors that exploits good capabilities of learning discriminative features with GAN models. In order to extract binary features we make use of intermediate layers of a GAN's discriminator [19]. To enforce good binary representation we incorporate two additional losses in training the discriminator: a distance matching regularizer that forces the propagation of distances from high-dimensional spaces to the low-dimensional compact space and an adjusted binarization representation entropy (BRE) regularizer [5] with weighted correlation.

## 3.1 GAN

The main idea of GAN [9] is based on game theory and assumes training of two competing networks: generator $G(\mathbf{z})$ and discriminator $D(\mathbf{x})$. The goal of GANs is to train generator $G$ to sample from the data distribution $p_{data}(\mathbf{x})$ by transforming the vector of noise $\mathbf{z}$ (of which, the prior is denoted as $p_{\mathbf{z}}(\mathbf{z})$). The discriminator $D$ is trained to distinguish the samples generated by $G$ from the samples from $p_{data}(\mathbf{x})$. The training problem formulation is as follows:

$$\min_{G} \max_{D} V(D, G) = \mathbb{E}_{\mathbf{x} \sim p_{data}(\mathbf{x})}[\log(D(\mathbf{x}))] + \mathbb{E}_{\mathbf{z} \sim p_{\mathbf{z}}(\mathbf{z})}[\log(1 - D(G(\mathbf{z})))]. \tag{1}$$

The model is usually trained with the gradient-based approaches by taking minibatch of fake images generated by transforming random vectors sampled from $p_{\mathbf{z}}(\mathbf{z})$ via the generator and minibatch of data samples from $p_{data}(\mathbf{x})$. They are used to maximize $V(D, G)$ with respect to parameters of $D$ by assuming a constant $G$, and then minimizing $V(D, G)$ with respect to parameters of $G$ by assuming a constant $D$.

However, to obtain more discriminative features on the intermediate layer of discriminator and stability of training process authors of [21] recommend, that generator $G$ should be trained using a *feature matching* procedure. The objective to train the generator $G$ is:

$$L_G = ||\mathbb{E}_{\mathbf{x} \sim p_{data}(\mathbf{x})} \mathbf{f}(\mathbf{x}) - \mathbb{E}_{\mathbf{z} \sim p_{\mathbf{z}}(\mathbf{z})} \mathbf{f}(G(\mathbf{z}))||_2^2, \tag{2}$$

where $\mathbf{f}(\mathbf{x})$ denotes the intermediate layer of the discriminator. In practical implementations it is usually the layer just before classification (*penultimate layer*).

Despite the fact that GANs are used for generating artificial examples from the data distribution, they can be also used as feature embeddings. This was initially discussed in [19] and further extended in [21], where the authors confirm that by incorporating a discriminator network, in a semi-supervised setting, they were able to obtain competitive results. There are a couple of benefits in using adversarial training for feature embeddings. First, during the adversarial training, the generator produces fake images with increasing quality and the discriminator is trained to distinguish between these and data examples. During this discriminative procedure the discriminator is forced to train more specific features that are characteristic for some regions of the feature space that are strongly associated with particular classes. Second, the adversarial training is done in an unsupervised setting without the need for tedious data annotation. Third, the feature matching approach (as in [21]) that is applied to train the discriminator results in generating fake images with similar feature characteristics, which forces the discriminator to extract more diverse features.

The most recent approaches for generating binary image descriptors aim at constructing binary vectors of low dimensionality. However, it was shown in [19] that the best performing representations in GANs can be obtained from high-dimensional intermediate layers of the discriminator. Therefore, in this work, we aim at transferring the Hamming distances from the high-dimensional space of intermediate layers to their binarized representations of low dimensionality to build our binary image descriptors. To that end, we propose a regularization technique that enforces this transfer, effectively leading to a construction of a compact yet discriminative binary descriptor. In Sec. 4.1 we define the layers used as high and low-dimensional representations for a given network architecture.

## 3.2 Distance Matching Regularizer

In this section we introduce a regularization loss function that aims at propagating the correlations between pairs of examples from high-dimensional space to low-dimensional representation, what is equivalent to propagating Hamming distances between two layers in the discriminator. We achieve this goal by taking a pair of vectors from two intermediate layers of the same network (discriminator) corresponding to two examples from a data batch and enforcing their binarized outputs to have similar normalized dot products.

Let $\mathbf{f}(\mathbf{x})$ and $\mathbf{h}(\mathbf{x})$ denote the low and high-dimensional intermediate layers of discriminator with the numbers of hidden units equal $K$ and $M$, respectively. We assume, that the number of hidden units for $\mathbf{f}(\mathbf{x})$ is significantly higher than the number of the units for $\mathbf{h}(\mathbf{x})$, $M \gg K$. The corresponding binary vectors $\mathbf{b}_f \in \{-1, 1\}^K$ and $\mathbf{b}_h \in \{-1, 1\}^M$ can be obtained using a sign function: $sign(a) = a/|a|$. The main problem with the sign activations is that they are not able to propagate the gradient backwards. In order to overcome this limitation we use the following quantization technique as in [5]: $softsign(a) = a/(|a| + \gamma)$, where $\gamma$ is a hyperparameter that is responsible for smoothing the $sign(\cdot)$ function. We define the vector $\mathbf{s}_f$ that is created by applying the $softsign(\cdot)$ function to each element of $\mathbf{f}(\mathbf{x})$: $s_{f,k} = softsign(f_k(\mathbf{x}))$.

Hamming distance between two binary vectors, $\mathbf{b}_1$ and $\mathbf{b}_2$ can be expressed using a dot product: $d_H(\mathbf{b}_1, \mathbf{b}_2) = -0.5 \cdot (\mathbf{b}_1^T \mathbf{b}_2 - M)$. As a consequence, distant vectors are characterized by low-valued dot products and close vectors are characterized by high values. Considering this property, we introduce the Distance Matching Regularizer (DMR) that aims at propagating the good coding properties of vectors $\mathbf{b}_h$ in high-dimensional space to the compact space of binary vectors $\mathbf{b}_f$ (represented by their soft proxy vectors $\mathbf{s}_f$). We define the DMR in the following manner:

$$L_{DMR} = \frac{1}{N(N-1)} \sum_{k,j=1,k \neq j}^{N} |\frac{\mathbf{b}_{h,k}^T \mathbf{b}_{h,j}}{M} - \frac{\mathbf{s}_{f,k}^T \mathbf{s}_{f,j}}{K}|, \qquad (3)$$

In terms of optimization procedure we assume constant values of high-dimensional vectors $\mathbf{b}_h$ and optimize the parameters of the discriminator with respect to $\mathbf{s}_f$. To make Hamming distances comparable between the high-dimensional and the low-dimensional spaces we normalize them by dividing by the corresponding vector dimensions.

The $L_{DMR}$ function can be interpreted as the empirical expected value of the loss function $l(d_h, d_f) = 2 \cdot |d_h - d_f|$, where $d_h$ is the normalized Hamming distance in high-dimensional space that is assumed to be constant and $d_f$ is the normalized distance in the low-dimensional space calculated on quantized vectors.

The motivation behind using this kind of regularization procedure is as follows. A usual approach for learning informative and discriminative feature embeddings is to take intermediate layers of the network, concatenate them and obtain high-dimensional representation that provides better benchmark results. However, practical applications such as image matching require binary, short and compact representations for sake of efficiency. Therefore, the role of $L_{DMR}$ regularizer is to map the good embeddings from high-dimensional to the compact binary space.

### 3.3 Adjusted Binarization Representation Entropy Regularizer

To increase the diversity of binary vectors in the low-dimensional layer we utilize BRE regularizer. It was initially applied in [5] to guide the discriminator $D$ to better allocate its model capacity, by encouraging the binary activation patterns on selected intermediate layers of $D$ to maximize the total entropy. To achieve this, floating-point features are binarized and the expected value of each of the binary dimensions is enforced to be equal to $0.0$[2] For that purpose the following regularizer is used:

$$L_{ME} = \frac{1}{K} \sum_{k=1}^{K} (\bar{s}_{f,k})^2, \tag{4}$$

where $\bar{s}_{f,k}$ are elements of $\bar{\mathbf{s}}_f = \frac{1}{N} \sum_{n}^{N} \mathbf{s}_{f,n}$ that represent the average of $N$ quantized binary vectors $\mathbf{s}_{f,n}$. To promote the independence between the binary variables, a loss term $L_{AC}$ is proposed in [5]:

$$L_{AC} = \frac{1}{N(N-1)} \sum_{k,j=1,k \neq j}^{N} \frac{|\mathbf{s}_{f,k}^T \cdot \mathbf{s}_{f,j}|}{K}. \tag{5}$$

The BRE regularizer introduced in [5] is defined as the sum of $L_{ME}$ and $L_{AC}$ losses. Effectively, we would like to increase the diversity of the binary vectors whose dot product is equal to zero, *i.e.* their distance is closer to the middle of the range, while for those vectors with dot product different then zero the importance of the diversity is lower, hence it can be downweighed. Therefore, we propose to amend the formulation of the BRE regularizer and replace $L_{AC}$ with its weighted version as defined below:

$$L_{MAC} = \sum_{k,j=1,k \neq j}^{N} \frac{\alpha_{k,j}}{Z} \frac{|\mathbf{s}_{f,k}^T \cdot \mathbf{s}_{f,j}|}{K}, \tag{6}$$

where weights $\alpha_{k,j}$ are associated with corresponding pairs $\mathbf{s}_{f,k}^T$ and $\mathbf{s}_{f,j}^T$ and $Z = \sum_{k,j=1,k \neq j}^{N} \alpha_{k,j}$ is normalization constant.

It can be observed that $p_{k,j} = \frac{\alpha_{k,j}}{Z}$ (for $\alpha_{k,j} \geq 0$) constitute the discrete distribution responsible for taking pairs of vectors for regularization. Practically, it was shown in [5] that $L_{AC}$ is the empirical estimation of $\mathbb{E}[\frac{\mathbf{b}^T \mathbf{b}'}{K}]$, where $\mathbf{b}$ and $\mathbf{b}'$ are zero-mean multivariate Bernoulli vectors that are independent. The $L_{MAC}$ criterion can be seen as empirical estimation of $\mathbb{E}_{p_{k,j}}[\frac{\mathbf{b}^T \mathbf{b}'}{K}]$ where the pairs $\mathbf{b}$ and $\mathbf{b}'$ are binded by the $p_{k,j}$ distribution.

We propose to define $\alpha_{k,j}$ in the following manner:

$$\alpha_{k,j} = \exp \left\{ \frac{-|\mathbf{b}_{h,k}^T \mathbf{b}_{h,j}|}{\beta \cdot M} \right\} = \exp \left\{ \frac{-|M - 2 \cdot d_H(\mathbf{b}_{h,k}, \mathbf{b}_{h,j})|}{\beta \cdot M} \right\}, \tag{7}$$

where $\mathbf{b}_{h,k}$ are binary vectors from the high-dimensional layer and $\beta$ is a hyperparameter that controls the variance of distances. As we mentioned before, we would like to promote low-dimensional vectors

for regularization that are not strongly correlated in high-dimensional space $h$, therefore we propose the function $\exp(-|a|/\beta)$ that takes the highest values for $a$ close to 0. As a consequence, we promote the pairs of vectors in the criterion $L_{MAC}$ for which distances are around $M/2$ and put less force to the pairs for which the distances in high-dimensional space are close to 0 and $M$. While optimizing $L_{MAC}$ in each iteration of gradient method we assume that $\mathbf{b}_{h,k}$ are constant and calculated from the $\mathbf{h(x)}$ layer of discriminator by application of the $sign(\cdot)$ function.

### 3.4  Training BinGAN

We train our BinGAN model in a typical unsupervised GAN scheme, by alternating procedure of updating the discriminator $D$ and generator $G$. The discriminator $D$ is trained using the following learning objective:

$$L = L_D + \lambda_{DMR} \cdot L_{DMR} + \lambda_{BRE} \cdot (L_{ME} + L_{MAC}) \qquad (8)$$

where $\lambda_{DMR}, \lambda_{BRE}$ are regularization parameters. $\lambda_{DMR}$ defines the impact of the DMR regularization term and $\lambda_{BRE}$ defines the impact of the two BRE terms, $L_{ME}$ and $L_{MAC}$. $L_D = -\mathbb{E}_{\mathbf{x} \sim p_{data}(\mathbf{x})}[\log(D(\mathbf{x}))] - \mathbb{E}_{z \sim p_{\mathbf{z}}(\mathbf{z})}[\log(1 - D(G(\mathbf{z})))]$ is the loss for training the discriminator.

The training procedure is performed in the standard methodology for this type of models assuming training generator $G$ and discriminator $D$ in alternating steps. The generator in BinGAN model is trained by minimizing the feature matching criterion provided by equation (2). The discriminator is updated to minimize the loss function that is defined by equation (8). The alternating procedure of updating generator and discriminator is repeated for each of the minibatches considered in the current epoch.

## 4  Results

We conduct experiments on two benchmark datasets, Brown gray-scale patches [3] and CIFAR-10 color images [12]. These benchmarks are used to evaluate the quality of our approach on image matching and image retrieval tasks, respectively.

### 4.1  Model Architecture and Parameter Settings

For both tasks, we use the same generator architecture and slight modifications of the discriminator. Below we outline the main features of both models and their parameters.

For the image matching task the discriminator is composed of 7 convolutional layers (3x3 kernels, 3 layers with 96 kernels and 4 layers with 128 kernels), two network-in-network (NiN) [15] layers (with 256 and 128 units respectively) and discriminative layer. For the low-dimensional feature space $\mathbf{b}_f$ we take the average-pooled NiN layer composed of 256 units. For the high-dimensional space $\mathbf{b}_h$ we take the reshaped output of the last convolutional layer that is composed of 9216 units.

For image retrieval the discriminator is composed of: 7 convolutional layers (3x3 kernels, 3 layers with 96 kernels and 4 layers with 192 kernels), two NiN layers with 192 units, one fully-connected layer with three variants of (16, 32, 64 units) and discriminative layer. For the low-dimensional feature space $\mathbf{b}_f$ we take fully-connected layer, and for the high-dimensional space $\mathbf{b}_h$ we take average-pooled last NiN layer.

There are 4 hyperparameters in our method: $\gamma$, $\beta$ and regularization parameters: $\lambda_{DMR}, \lambda_{BRE}$. In all our experiments, we fix the parameters to: $\lambda_{DMR} = 0.05$, $\lambda_{BRE} = 0.01$, $\gamma = 0.001$ and $\beta = 0.5$.

The values of the hyperparameters were set according to the following motivations. The hyperparameter $\gamma$ controls the softness of the $sign(\cdot)$ function and the value was set according to suggestions provided in [5] therefore additional tuning was not needed. The value of a scaling parameter $\beta$ was set according to prior assumptions based on the analysis of the impact of scaling factor for the Laplace distribution. We scale the distances by the number of the units (M), therefore the value of $\beta$ can be constant among various applications. The values of regularization terms $\lambda_{BRE}$ and $\lambda_{DMR}$ were fixed empirically following the methodology provided in [6].

| Method | 16 bit | 32 bit | 64 bit |
|--------|--------|--------|--------|
| KHM | 13.59 | 13.93 | 14.46 |
| SphH | 13.98 | 14.58 | 15.38 |
| SpeH | 12.55 | 12.42 | 12.56 |
| SH | 12.95 | 14.09 | 13.89 |
| PCAH | 12.91 | 12.60 | 12.10 |
| LSH | 12.55 | 13.76 | 15.07 |
| PCA-ITQ | 15.67 | 16.20 | 16.64 |
| DH | 16.17 | 16.62 | 16.96 |
| DeepBit | 19.43 | 24.86 | 27.73 |
| DBD-MQ | 21.53 | 26.50 | 31.85 |
| **BinGAN** | **30.05** | **34.65** | **36.77** |

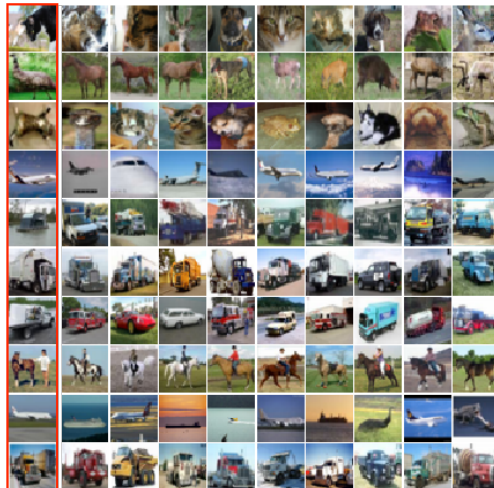

Figure 1: **(Left)** Performance comparison (mAP, %) of different unsupervised hashing algorithms on the CIFAR-10 dataset. This table shows the mean Average Precision (mAP) of top 1000 returned images with respect to different number of hash bits. We report the results for all the methods except for BinGAN after [6]. **(Right)** Top retrieved image matches from CIFAR-10 dataset for given query images from test set - first column.

## 4.2 Image Retrieval

In this experiment we use CIFAR-10 dataset to evaluate the quality of our approach in image retrieval. CIFAR-10 dataset has 10 categories and each of them is composed of 6,000 pictures with a resolution $32 \times 32$ color images. The whole dataset has 50,000 training and 10,000 testing images.

To compare the binary descriptor generated with our BinGAN model with the competing approaches, we evaluate several unsupervised state-of-the methods, such as: KMH [10], Spherical Hashing (SphH)[11], PCAH [29], Spectral Hashing (SpeH)[30], Semantic Hashing (SH) [20], LSH [2], PCT-ITQ [8], Deep Hashing (DH)[16], DeepBit[14], deep binary descriptor with multiquantization (DBD-MQ)[6]. For all methods except DH, DeepBit, DBD-MQ and ours, we follow [16] and compute hashes on 512-d GIST descriptors. The table in Fig. 1 shows the CIFAR10 retrieval results based on the mean Average Precision (mAP) of the top 1000 returned images with respect to different bit lengths. Fig. 1 shows top 10 images retrieved from a database for given query image from our test data.

Our method outperforms DBD-MQ method, the unsupervised method previously reporting state-of-the-art results on this dataset, for 16, 32 and 64 bits. The performance improvement in terms of mean Average Precision reaches over 40%, 31% and 15%, respectively. The most significant performance boost can be observed for the shortest binary strings, as thanks to the loss terms introduced in our method, we explicitly model the distribution of the information in a low-dimensional binary space.

## 4.3 Image Matching

To evaluate the performance of our approach on image matching task, we use the Brown dataset [3] and train binary local feature descriptors using our BinGAN method and competing previous methods, applying the methodology described in [14]. The Brown dataset is composed of three subsets of patches: Yosemite, Liberty and Notredame. The resolution of the patches is $64 \times 64$, although we subsample them to $32 \times 32$ to increase the processing efficiency and use the method to create binary descriptors in practice. The data is split into training and test sets according to the provided ground truth, with 50,000 training pairs (25,000 matched and 25,000 non-matched pairs) and 10,000 test pairs (5,000 matched, and 5,000 non-matched pairs), respectively.

Tab. 1 shows the false positive rates at $95\%$ true positives (FPR@95%) for binary descriptors generated with our BinGAN approach compared with several state-of-the-art descriptors. Among the compared approaches, Boosted SSC [22], BRISK [13], BRIEF [4], DeepBit [14] and DBD-MQ

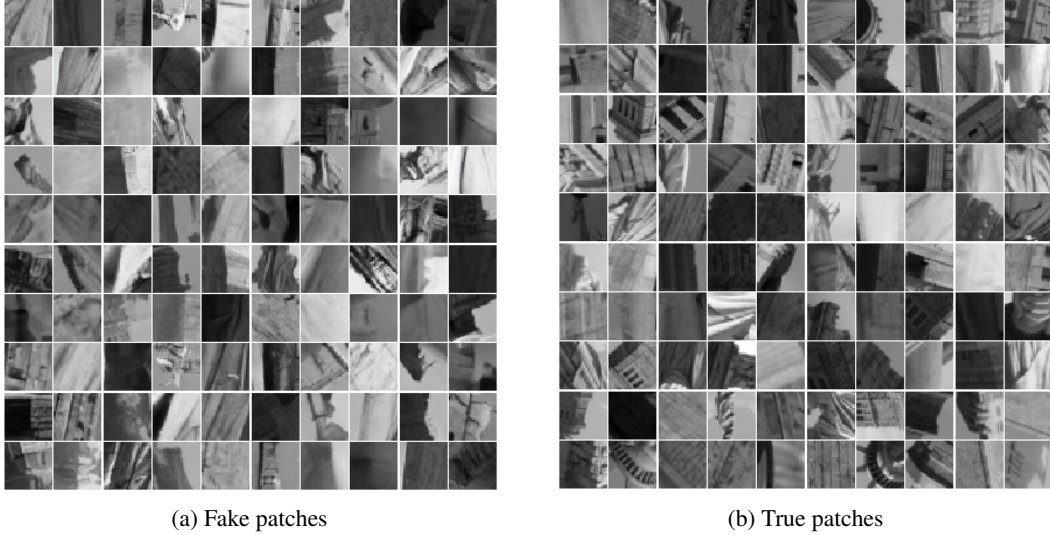

| (a) Fake patches | (b) True patches |

Figure 2: We present generative capabilities of BinGAN model for the Liberty dataset. Fake patches generated by the model are shown in Fig. 2a and true patches from the data in Fig. 2b.

Table 1: False positive rates at $95\%$ true positives (FPR@95%) obtained for our BinGAN descriptor compared with the state-of-the-art binary descriptors on Brown dataset ($\%$). Real-valued SIFT features are provided for reference. We report all the results from [6], except for L2-Net and BinGAN.

| Train | Yosemite | | Notre Dame | | Liberty | | Average |
|---|---|---|---|---|---|---|---|
| Test | Notre Dame | Liberty | Yosemite | Liberty | Notre Dame | Yosemite | FPR@95% |
| **Supervised** | | | | | | | |
| LDAHash (16 bytes) | 51.58 | 49.66 | 52.95 | 49.66 | 51.58 | 52.95 | 51.40 |
| D-BRIEF (4 bytes) | 43.96 | 53.39 | 46.22 | 51.30 | 43.10 | 47.29 | 47.54 |
| BinBoost (8 bytes) | 14.54 | 21.67 | 18.96 | 20.49 | 16.90 | 22.88 | 19.24 |
| RFD (50-70 bytes) | 11.68 | 19.40 | 14.50 | 19.35 | 13.23 | 16.99 | 15.86 |
| Binary L2-Net [26] (32 bytes) | 2.51 | 6.65 | 4.04 | 4.01 | 1.9 | 5.61 | 4.12 |
| **Unsupervised** | | | | | | | |
| SIFT (128 bytes) | 28.09 | 36.27 | 29.15 | 36.27 | 28.09 | 29.15 | 31.17 |
| BRISK (64 bytes) | 74.88 | 79.36 | 73.21 | 79.36 | 74.88 | 73.21 | 75.81 |
| BRIEF (32 bytes) | 54.57 | 59.15 | 54.96 | 59.15 | 54.57 | 54.96 | 56.23 |
| DeepBit (32 bytes) | 29.60 | 34.41 | 63.68 | 32.06 | 26.66 | 57.61 | 40.67 |
| DBD-MQ (32 bytes) | 27.20 | 33.11 | 57.24 | 31.10 | **25.78** | 57.15 | 38.59 |
| BinGAN (32 bytes) | **16.88** | **26.08** | **40.80** | **25.76** | 27.84 | **47.64** | **30.76** |

Table 2: Ablation study. False positive rates at $95\%$ true positives (FPR@95%) for three settings of $\lambda$ parameters when training BinGAN for image matching. Optimizing all three loss terms leads to the best performance on the Brown dataset.

| Train | Yosemite | | Notre Dame | | Liberty | | Average |
|---|---|---|---|---|---|---|---|
| Test | Notre Dame | Liberty | Yosemite | Liberty | Notre Dame | Yosemite | FPR@95% |
| $\lambda_{DMR} = \lambda_{BRE} = 0$ | 32.72 | 39.44 | **39.44** | 27.92 | 27.24 | 50.48 | 36.21 |
| $\lambda_{DMR} = 0 \quad \lambda_{BRE} = \mathbf{0.01}$ | 30.12 | 36.28 | 44.2 | **24.28** | **26.44** | 51.88 | 35.53 |
| $\lambda_{DMR} = \mathbf{0.05} \quad \lambda_{BRE} = 0$ | 24.68 | 26.96 | 40.16 | 27.00 | 27.28 | **45.28** | 31.90 |
| $\lambda_{DMR} = \mathbf{0.05} \quad \lambda_{BRE} = \mathbf{0.01}$ | **16.88** | **26.08** | 40.80 | 25.76 | 27.84 | 47.64 | **30.76** |

[6] are unsupervised binary descriptors while LDAHash [25], D-BRIEF [28], BinBoost [27] and RFD [7] are supervised. The real-valued SIFT [17] is provided for reference. Our BinGAN approach achieves the lowest FPR@95% value of all unsupervised binary descriptors. The improvement over the state-of-the-art competitor, DBD-MQ, is especially visible when testing on Yosemite.

Furthermore, we examine the influence of BinGAN's regularization terms on the performance of the resulting binary descriptor. Tab. 2) shows the results of this ablation study. Using binarized features from a GAN trained without any additional loss terms provides state-of-the-art results in terms of average FPR@95%. By adding Distance Matching Regularizer ($\lambda_{DMR} \neq 0$) we can observe significant improvement for almost all testing cases. Additional performance boost can be observed

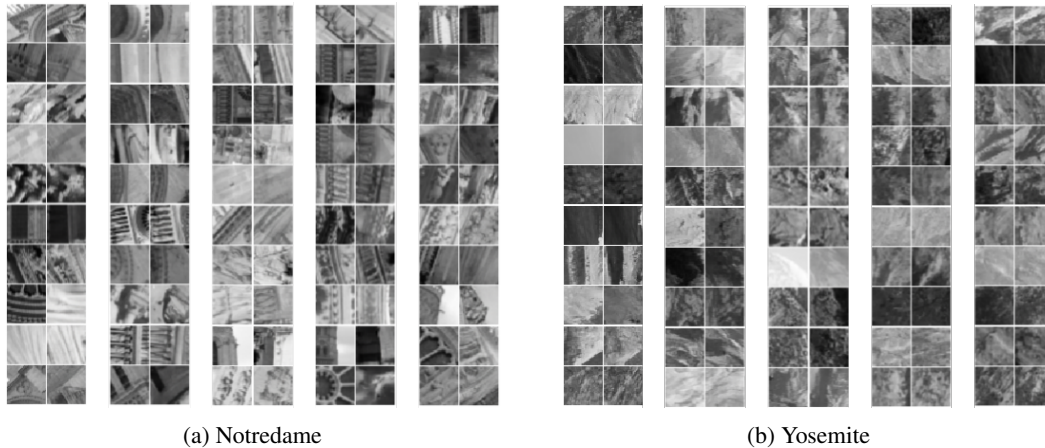

|              (a) Notredame              |              (b) Yosemite              |

Figure 3: The set of randomly selected patches from the original data (odd columns) and corresponding synthetically generated patches (even columns) that are at the closest Hamming distance to the true patch in the binary descriptor space.

when adding the adjusted BRE regularizer. We can therefore conclude that the results of our BinGAN approach can be attributed to a combination of two regularization terms proposed in this work.

### 4.4  Generative Capabilities of BinGAN

Contrary to previous methods for learning binary image descriptors, our approach allows to synthetically generate new image patches that can be then used for semi-supervised learning. Fig. 2 presents the images created by a generator trained on the Liberty dataset. Fake and true images are difficult to differentiate. Additionally, Fig. 3 presents patch pairs that consist of a true patch and a synthetically generated patch with the closest Hamming distance to the true patch in the binary descriptor space. The majority of generated patches are fairly similar to the original ones, which can hint that those patches can be used for semi-supervised training of more powerful binary descriptors, although this remains our future work.

## 5  Conclusions

In this work, we presented a novel approach for learning compact binary image descriptors that exploit regularized Generative Adversarial Networks. The proposed BinGAN architecture is trained with two regularization terms that enable weighting the importance of dimensions with the correlation matrix and propagate the distances between high-dimensional and low-dimensional spaces of the discriminator. The resulting binary descriptor is highly compact yet discriminative, providing state-of-the-art results on two benchmark datasets for image matching and image retrieval.

## Acknowledgements

This research was partially supported by the Polish National Science Centre grant no. UMO-2016/21/D/ST6/01946 as well as Google Sponsor Research Agreement under the project "Efficient visual localization on mobile devices".

The research conducted by Maciej Zieba has been partially co- financed by the Ministry of Science and Higher Education, Republic of Poland.

We would especially link to thank Karol Kurach, Jan Hosang, Adam Bielski and Aleksander Holynski for their valuable insights and discussion.

## Footnotes

[1]The code is available at: `github.com/maciejzieba/binGAN`

[2]We assume $\{-1, 1\}$ and independence between them is enforced by minimizing the correlation.

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
