[Reviews · NeurIPS 2018]

Reviewer 1



This paper proposes a novel GAN formulation that allows a GAN to learn discriminative and compact binary codes for image patches. Using (hand designed) binary descriptors has quite some tradition in computer vision. Binary descriptors have several attractive properties, eg. compactness, fast implementation etc. Using GANs for this purpose is not new (as nicely explained in the paper). The novelty of the method is that it employs two regularization terms that allow weighting of the importance of dimensions with the correlation and to propagate the distances between high and low dim. spaces. This leads to good results and compact codes. So overall this is minor but importnat contribution. Overall the paper is well written and has a clear contribution. My main concerns are that the paper does not explain how to set the hyperparameters. In addition, the experiments are performed on moderately sized databases. An additional large-scale experiment on eg. one of the huge databases for city-scale reconstruction would show how the method scales.

Reviewer 2



This paper proposes to convert the intermediate layer of the GAN discriminator to low dimensional binary representation, which then can be used for retrieval. Direct quantization of the low dimensional layer is not very informative. In order to preserve the distance information between sample pairs, the author proposes to transfer the distance information from a higher dimensional layer. Quality and clarity: The paper is well written and easy to follow. It is also easy to reimplement given the math expressions. The author explains the motivation of the regularization terms and the improvements over previous works. The experiments clearly shows that the proposed method is better than previous methods. (I'm not very familiar with relevant literature, i.e. how hard it is to beat sota in these tasks.) Originality and Significance: This method is based on the GAN framework to do unsupervised training, the originality comes from the regularizations proposed in this paper. It is original in this sense. Despite the good performance shown, the regularizations proposed in this paper are combination of empirical losses. The fact that each of the terms needs a hyperparam makes it relatively unappealing. The theoretical value and possible extension is relatively weak.

Reviewer 3



Summary This paper proposes a variant of GAN to learn compact binary descriptors for image patch matching. The authors introduce two novel regularizers to propagate Hamming distance between two layers in the discriminator and encourage the diversity of learned descriptors. The presentation is easy to follow, and the method is validated by benchmark datasets. Major concerns: [Motivation and Presentation] First of all, it’s not so clear the reason why adversarial training helps to learn compact binary descriptors. In addition, the motivation on DMR is also not fully addressed in my sense. In my understanding, the discriminator has two binary representation layers; one of them has the larger number of bits and the other is used for the compact binary descriptor. However, it’s not clearly described in the current draft. [About Experiments] There exists a simple baseline by just combining state-of-the-art image descriptors with binary embedding algorithms. Specifically, in stead of learning descriptors and its binary representation together, let us split two parts and use SOTA algorithms in both directions. For example, what about using the following algorithm followed by deep hashing to generate compact binary descriptors? - Image Patch Matching Using Convolutional Descriptors with Euclidean Distance, ACCV’16 workshop, https://arxiv.org/abs/1710.11359 I believe that there exist more follow-up works to learn more meaningful descriptors for matching image patches. Preliminary rating: borderline This paper has some potential to be a good application of GAN, which learns compact binary descriptors for image matching. Yet, the motivation on network architecture and experiments are not sufficient. Please elaborate them in the rebuttal, if possible. == Updated review == The rebuttal addressed my concerns on the experiment results. Yet, I still think that the technical novelty on the proposed methods is not strong for NIPS. Specifically, the entropy regularizer has been studied to learn decorrelated hash bits before; - SPEC Hashing: Similarity Preserving algorithm for Entropy-based Coding, CVPR’10. Though the exact form used in this manuscript might not been employed before, it is quite common to learn decorrelated bits by maximizing entropy on them. Moreover, the other regularizer (DMR) is designed to reduce the dimensionality of high-dimensional binary codes learned from an intermediate layer of discriminator. It’s a very natural and practical regularizer, but I’m not sure that it’s solely enough to be a nips paper. Therefore, I haven’t changed my original score (5).